# Real-Time Semantic Understanding and Segmentation of Urban Scenes for Vehicle Visual Sensors by Optimized DCNN Algorithm

**Yanyi Li** [1] , **Jian Shi** [2,*] **and Yuping Li** [1]

1   College of Surveying and Geo-Informatics, Tongji University, Shanghai 200092, China;
   2131939@tongji.edu.cn (Y.L.); 2131946@tongji.edu.cn (Y.L.)
2   School of Electronic Information Engineering, Shanghai Dianji University, Shanghai 201306, China
*   Correspondence: 201801020123@sdust.edu.cn; Tel.: +86-191-1714-8462

**Abstract:** The modern urban environment is becoming more and more complex. In helping us identify surrounding objects, vehicle vision sensors rely more on the semantic segmentation ability of deep learning networks. The performance of a semantic segmentation network is essential. This factor will directly affect the comprehensive level of driving assistance technology in road environment perception. However, the existing semantic segmentation network has a redundant structure, many parameters, and low operational efficiency. Therefore, to reduce the complexity of the network and reduce the number of parameters to improve the network efficiency, based on the deep learning (DL) theory, a method for efficient image semantic segmentation using Deep Convolutional Neural Network (DCNN) is deeply studied. First, the theoretical basis of the convolutional neural network (CNN) is briefly introduced, and the real-time semantic segmentation technology of urban scenes based on DCNN is recommended in detail. Second, the atrous convolution algorithm and the multi-scale parallel atrous spatial pyramid model are introduced. On the basis of this, an Efficient Symmetric Network (ESNet) of real-time semantic segmentation model for autonomous driving scenarios is proposed. The experimental results show that: (1) On the Cityscapes dataset, the ESNet structure achieves 70.7% segmentation accuracy for the 19 semantic categories set, and 87.4% for the seven large grouping categories. Compared with other algorithms, the accuracy has increased to varying degrees. (2) On the CamVid dataset, compared with segmentation networks of multiple lightweight real-time images, the parameters of the ESNet model are around 1.2 m, the highest FPS value is around 90 Hz, and the highest mIOU value is around 70%. In seven semantic categories, the segmentation accuracy of the ESNet model is the highest at around 98%. From this, we found that the ESNet significantly improves segmentation accuracy while maintaining faster forward inference speed. Overall, the research not only provides technical support for the development of real-time semantic understanding and segmentation of DCNN algorithms but also contributes to the development of artificial intelligence technology.

**Keywords:** deep learning; image semantic segmentation; deep convolutional neural networks; efficient symmetric network; segmentation method





## 1. Introduction

In recent years, with the rapid development of high-speed mobile communication technology and computer software and hardware technology, autonomous unmanned systems represented by unmanned driving and mobile robots have attracted a new wave of development. Compared with the unmanned driving technology at the end of the 20th century, the existing unmanned driving has made great progress. However, there is still a certain gap compared with the standards stipulated by the Society of Automotive Engineers (SAE). When faced with a variety of driving environments, unmanned systems can adapt as human beings [1,2]. Humans mainly obtain information through vision. Therefore, the core task of realizing universal unmanned driving is to enable the unmanned system to

understand the traffic scene similarly to a human. Computer vision (CV), a technology focused on making computers understand images similarly to humans, has received a lot of attention. Limited by the theoretical level and computing resources, the early CV technology only tried low-level image understanding such as image segmentation and edge detection, and there is no obvious boundary between it and classical image processing [3]. However, with the rapid expansion of society, especially in the construction of smart cities, the requirements for intelligent image recognition technology are becoming higher and higher. Although the application of intelligent image recognition technology in the current society is not mature enough, many studies have provided technical support for it.

CV is gradually subdivided into various sub-tasks such as semantic segmentation, object detection, and image classification according to the needs of understanding in different scenarios. In recent years, with the rise of deep learning (DL), the use of Deep Convolutional Neural Networks (DCNN) to achieve automatic learning of image features has become a consensus in the field of CV, and the field of semantic segmentation has also entered the fast lane of development [4,5]. However, large labeled datasets such as multispectral imagery (MSI) for other sensor modalities cannot be obtained due to the large cost and labor required. Kemker et al. (2018) [6] adopted the most advanced DCNN framework in CV for semantic segmentation of MSI. To overcome the label scarcity of MSI data, the generated synthetic MSI was replaced with real MSI to initialize the DCNN framework. Zhang et al. (2020) [7] designed a complete symmetric network structure including a pool index and a convolution for fusing semantic information and image features. The bottleneck layer is constructed with $1 \times 1$ convolution to extract details, reduce the number of parameters, deepen the filtering depth, build an end-to-end semantic segmentation network, and improve the activation function to further improve the network performance. Garg et al. (2021) [8] proposed an automatic machine learning (ML) method and system for semantic segmentation of remote sensing images, which belongs to the field of artificial intelligence (AI). The semantic segmentation algorithm of remote sensing images based on multiple remote sensing imaging indices improves the segmentation accuracy through background loss optimization and imaging index calculation. Through self-migration fine-tuning training method and hyperparameter optimization method, the hyperparameter optimization of the DCNN for a single machine with multiple image processors can improve the segmentation accuracy and optimization efficiency at the same time. The search space for parameter sharing is set, and the multi-scale information of the DCNN is extracted through the spatial pyramid pooling module with the search strategy based on the policy gradient to search for the optimal internal network structure of the DCNN and improve the efficiency of semantic segmentation and classification accuracy of remote sensing images. Sharifi (2020) [9] briefly analyzed the application of ML in CV processing, introduced the application progress of ML in the field of image detection and semantic segmentation, and emphatically analyzed the algorithm principle of the typical classification algorithm-random forest. At the end, the application prospect of ML in CV has been prospected. Sharifi (2021) [10] designed an end-to-end depth-supervised fully convolutional network based on convolutional neural networks (CNN) to solve the problem of automatic segmentation of brain tissue in brain images. For 3D brain images, the data are first cut into 2D image slices. On the basis of supervising the fully convolutional network structure, a deep supervision mechanism is added; that is, the loss value feedback is obtained in advance in the multi-level structure of feature extraction. By introducing a deep supervision structure, the improved model achieves a more accurate segmentation effect in the segmentation task of brain tissue. There are six most commonly used segmentation networks, which are semantic Segmentation Network (SegNet), Efficient Segmentation Network (ENet), Efficient Spatial Pyramid Network (ESPNet), Cross Guidance Network (CGNet), Efficient Residual Factorized Conv Network (ERFNet), and Image Cascade Network (ICNet). In addition, the accuracy of the above semantic segmentation is also high [11–13]. To sum up, there is still a certain gap between the accuracy of real-time semantic segmentation models and high-precision

models. Among them, the most obvious gap is the segmentation of small objects. In urban traffic scenes, small objects such as signs and traffic lights are significant for the understanding of traffic scenes. Therefore, how to ensure segmentation accuracy is still the main challenge facing the current semantic segmentation.

Based on the existing deep learning network, this study proposes a more efficient and accurate semantic segmentation network, which can better deal with the problem of semantic segmentation in urban environment. We have carried out experimental tests on a variety of common ground objects in the city and found that the ESNet can better complete the task of semantic segmentation. As the modern urban environment becomes more and more complex, in the process of helping us accurately understand the surrounding environment and identify surrounding objects, mobile terminals on vehicles also rely more on the semantic segmentation ability of deep learning networks. The network structure proposed in ESNet not only considers accuracy and efficiency, but also completes the semantic segmentation of scene elements in the city, which will play an important role in the research of urban environment perception and automatic driving.

The content structure of the article is as follows. First, introducing the atrous spatial pyramid model and atrous convolution algorithm, an Efficient Symmetric Network (ESNet) of semantic segmentation model for autonomous driving scenes is constructed. Then, the segmentation performance of the model is evaluated on the urban scene dataset. On account of this, the technical principle of an efficient symmetric real-time semantic segmentation network based on DCNN is first expounded. After that, its application design concept in urban traffic is discussed. In the end, the simulation test is carried out by a computer, which comprehensively evaluates the designed ESNet model. The innovation is to apply DL-based image semantic segmentation technology to the smart transportation system to promote the development of urban traffic management. The research not only provides technical support for the development of semantic segmentation but also contributes to the development of intelligent transportation.

## 2. An ESNet of Real-Time Semantic Segmentation Based on DCNN

### 2.1. Basic Principles of CNN

The LeNet-5 network proposed by LeChen can effectively recognize handwritten digits, bring the CNN back into people's view, and lay the basic structure of the CNN. Additionally, CNN is a branch of deep neural network (DNN). The diagram of the network is shown in Figure 1.

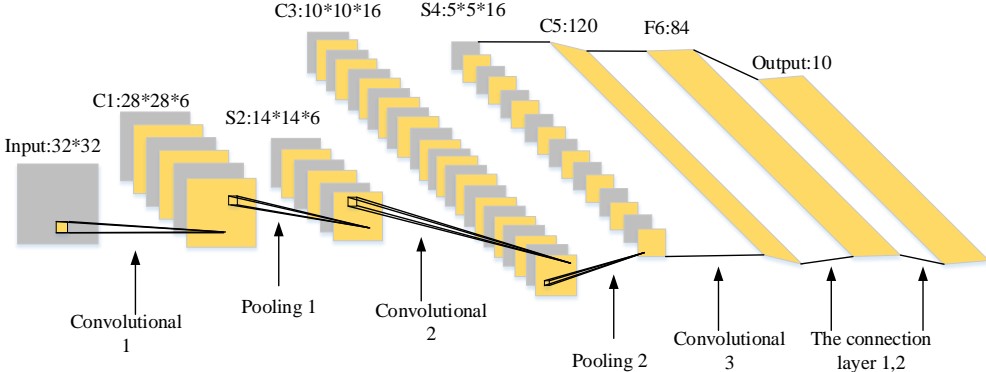

**Figure 1.** Architecture of LeNet.

(1) Convolutional layer

The core layer in CNN is the convolutional layer, which contains multiple convolution kernels. Different convolution kernels can extract the local features of the images in the input network. In general, low-level convolutional layers can only extract low-level local features such as corners and edges. In contrast, the extraction of high-level feature representations requires high-level convolutional layers [14–16]. The number of convolution

kernels, the step size of convolution, the size of the convolution kernel, and the padding are the main parameters of the convolutional layer. The size of the convolution kernel is set to K × K, the step size of convolution is set to 1, and no padding. The obtained convolution calculation is shown in Equation (1).

$$c_{i,j}^s = \sum_{m=0}^{K} \sum_{n=0}^{K} \sum_{m=0}^{K} W_{m,n}^s X_{m+i,n+j}^s + b^s \tag{1}$$

In Equation (1):

$X$ is the single-channel feature map of the input network.

$C(i, j)$ is the element of the $i$-th row and the $j$-th column of the feature map output from the network structure.

$W$ is the weight value.

$b$ is the bias value.

$S$ is the different biases and weights of different convolution kernels.

The process of convolution operation is shown in Figure 2.

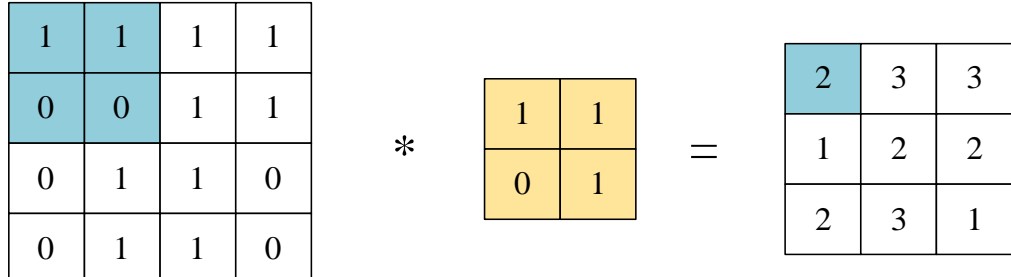

**Figure 2.** The process of the convolution operation.

(2) Pooling layer

The pooling layer, also known as the down-sampled layer, reduces the dimension of the feature image by down-sampling. The redundant feature information is reduced, and the dimension reduction of the data is realized. In this process, the quality of the picture will not be affected. The pooling layer can be divided into average pooling and maximum pooling. Average pooling refers to dividing the input feature image into multiple regions and calculating the average value as the output. Max pooling takes the maximum value in each region as the output [17,18]. The diagram of the two pooling operations is shown in Figure 3.

(3) Fully connected layer

The fully connected layer is composed of neurons on a single layer, and its neuron nodes are connected to the neuron nodes of the previous layer. The fully connected layer is usually stacked behind the convolutional layer in the CNN, and the neural network (NN) can contain multiple fully connected layers to integrate the extracted feature information. Suppose the role of the pooling layer and the convolutional layer is to map the original data to the feature space of the hidden layer. In that case, the role of the fully connected layer is to map the learned feature representation to the sample label space. It is equivalent to the "classifier" in the entire CNN [19,20].

(4) Activation function

The activation function plays a vital role in the composition of the NN. If there is no activation function in the NN, it is no different from a single-layer network and can only perform some simple linear calculations. The activation function can make the network more flexible to adjust the mapping function, and the function is often nonlinear such that the DNN can fit any complex function [21–23]. Common activation functions are Sigmoid activation function, Tanh activation function, Relu activation function, Leaky Relu activation function. The plot of these activation functions is shown in Figure 4.

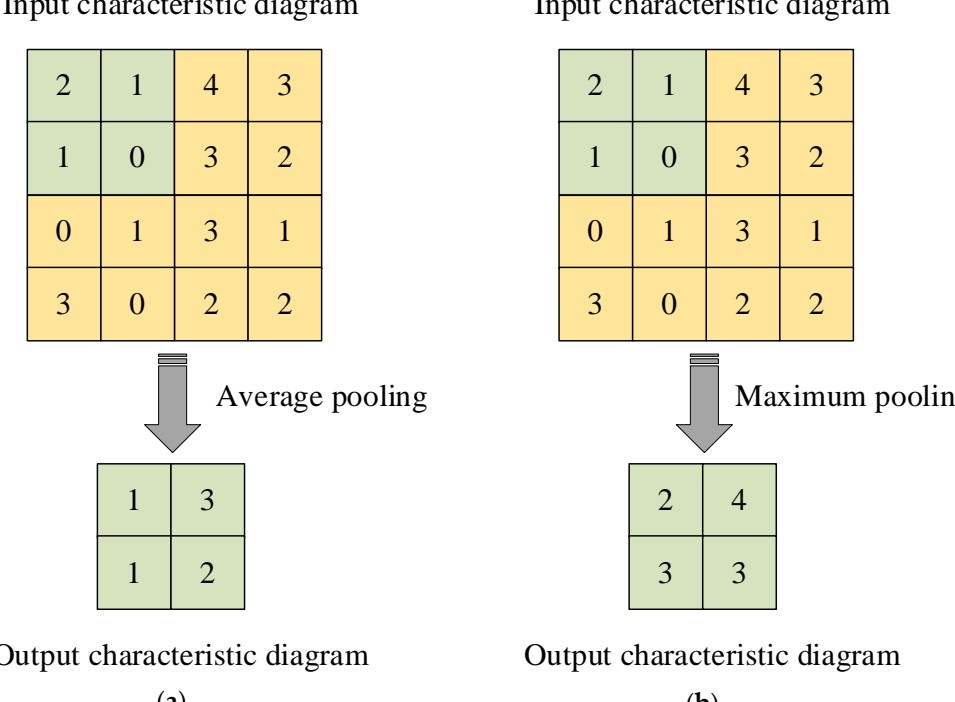

**Figure 3.** The diagram of the two pooling operations: (**a**) average pooling; (**b**) maximum pooling.

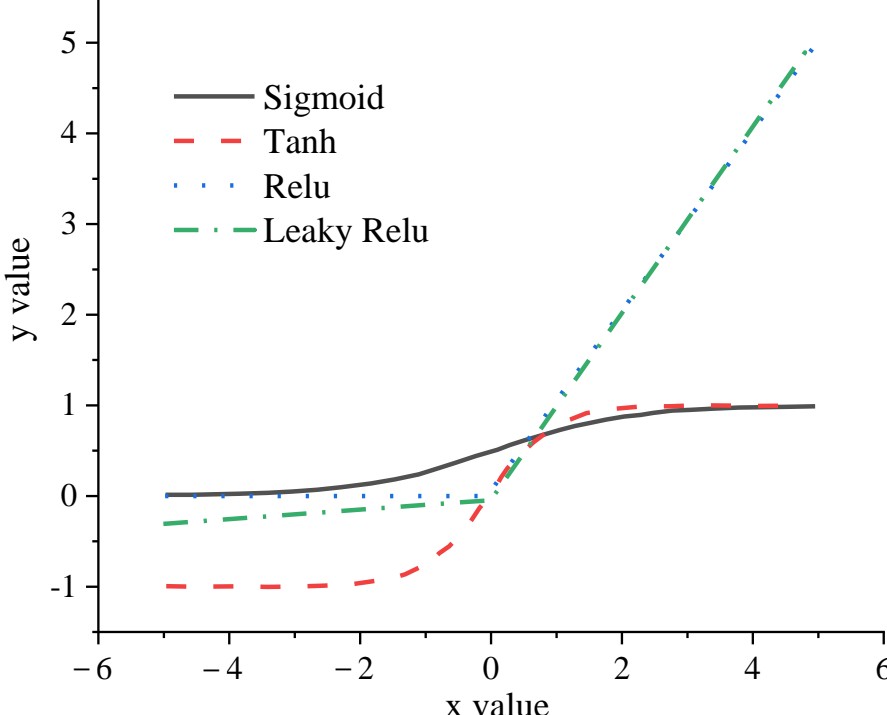

**Figure 4.** The plots of the activation functions.

The most commonly used activation function in the early classification network is the Sigmoid activation function, and its expression is shown in Equation (2).

$$f(x) = \frac{1}{1 + e^{-x}} \tag{2}$$

The Sigmoid function maps the input data between 0 and 1. When the input data are less than 0 and the value is small, the output will be close to 0; when the input data are greater than 0 and the value is large, the output will be close to 1.

The image curve structure of the Tanh activation function is similar to that of the Sigmoid function. However, the Tanh function will map the input data between −1 and 1, and the average of the output results is 0, which can speed up the convergence of the network to a certain extent. When the input value is small or large, the vanishing gradient problem still occurs, resulting in stagnation of training. The expression of the Tanh activation function is shown in Equation (3).

$$f(x) = \frac{e^x - e^{-x}}{e^x + e^{-x}} \tag{3}$$

In recent years, the Sigmoid activation function has been gradually replaced by the Relu function and has been widely used in various networks. The Relu activation function has two main advantages. One is to effectively solve the vanishing gradient of the network caused by the Tanh and Sigmoid functions. When the input data are positive, the gradient can always be guaranteed to be 1. The other is compared with the Sigmoid and Tanh function, the convergence efficiency of the Relu activation function has been significantly improved [24,25]. The expression of this function is shown in Equation (4).

$$f(x) = \begin{cases} x, \ x \geq 0 \\ 0, \ x < 0 \end{cases} \tag{4}$$

It can also be expressed by Equation (5).

$$f(x) = \max(0, \ x) \tag{5}$$

The Leaky Relu function image structure is the same as that of the Relu function. This function can effectively solve the problem of vanishing gradient while speeding up the training efficiency of the network. The expression of the Leaky Relu function is shown in Equation (6).

$$f(x) = \begin{cases} x, \ x \geq 0 \\ \alpha x, \ x < 0 \end{cases}(\alpha > 0) \tag{6}$$

It can also be expressed by Equation (7).

$$f(x) = \max(\alpha x, \ x) \tag{7}$$

### 2.2. Evaluation Indicators of Semantic Segmentation

As one of the most critical topics in CV, semantic segmentation aims to segment different objects in pictures and identify their semantic categories. Early semantic segmentation mostly follows a segmentation-first-classification scheme. With the development of DL, semantic segmentation is mainly transformed into a pixel-level classification task, semantic classification of pixels based on image patches around them. To measure the accuracy of semantic segmentation, commonly used evaluation indicators include Pixel Accuary (PACC) and Intersection over Union (IoU).

The calculation of the accuracy of pixel classification is the proportion of correctly identified pixels to all pixels, and its mathematical expression is shown in Equation (8).

$$P - ACC = \frac{\sum_i^N \delta\left(\hat{l}_l, \ l_i\right)}{N} \tag{8}$$

$N$ is the total number of pixels in the test set, $\hat{l}_l, l_i$ refers to the predicted label and true value of pixel $i$, respectively, and $\delta$ is the indicator function shown in Equation (9).

$$\delta(x, \ y) = \begin{cases} 1 \text{ if } x = y \\ 0 \text{ otherwise} \end{cases} \tag{9}$$

The most commonly used evaluation criterion for semantic segmentation is the IoU. The IoU is a common criterion for evaluating the similarity of sets A and B in statistics. This criterion calculates the ratio of the number of elements in the intersection of A and B to the number of elements in the union. The higher the value, the more similar the two sets are. When it is extended to the field of semantic segmentation, the set involved in the calculation of the IoU of a certain class $(k)$ is the set of pixels predicted to be this class and the set whose true value is this class, and its mathematical expression is shown in Equation (10).

$$\text{IoU}(k) \ = \ \frac{\left( \sum_i^N \delta(l_i,\ k) \ \& \ \delta\left(\hat{l}_l,\ k\right) \right)}{\left( \sum_i^N \delta(l_i,\ k) \mid \delta\left(\hat{l}_l,\ k\right) \right)} \tag{10}$$

In Equation (10), the meaning of each symbol is consistent with that of Equation (8). Frequency-weighted Intersection over Union (FwIoU) is an extension of the average IoU. The pixel categories of the predicted target are weighted separately, and the weight is determined by the frequency of each pixel in all target pixels. Its calculation equation is shown in Equation (11).

$$FwIoU = \frac{1}{\sum_{i=0}^{k}\sum_{j=0}^{k} p_{ij}} \sum_{j=0}^{k} \frac{\sum_{j=0}^{k} p_{ij}p_{ii}}{\sum_{j=0}^{k} p_{ij} + \sum_{j=0}^{k} p_{ji} - p_{ii}} \tag{11}$$

Computational complexity is usually assessed using two main indicators, one is used to measure the speed of the model's forward prediction, and the other is considered from the perspective of the model's computational memory or storage.

Forward inference time calculates the entire processing time of a single image input into an algorithm or the output result of pixel-level semantic segmentation from the system to the network. This indicator is affected mainly by the hardware used; thus, an evaluation of this indicator for any algorithm should specify the hardware used.

Memory usage is important for semantic segmentation tasks, especially when deploying semantic segmentation algorithms on some resource-constrained devices, such as mobile phones, digital cameras, or safety systems in autonomous driving; they need to be carefully evaluated. Memory usage may change significantly during its operation for a complex semantic segmentation network. Therefore, peak memory is generally used, which demonstrates the maximum memory required for a single image when the model is running.

### 2.3. Analysis of the Principle of Atrous Convolution

(1)  Receptive field

The receptive field refers to the area of the original input image perceived by the pixels on the output feature image of a particular convolutional layer. It plays a significant role in the structure of DCNN. The larger the receptive field, the more global information contained in the NN features. The receptive field of DCNN can be enlarged in two ways:

One is to expand the receptive field of the entire network structure through down-sampled strategies such as the confluence layer with step size and convolutional layer; the other is to use the atrous convolution to expand the receptive field NN. This method can keep the image's resolution unchanged [26,27]. The specific analysis of the calculation process of the receptive field is shown in Figure 5.

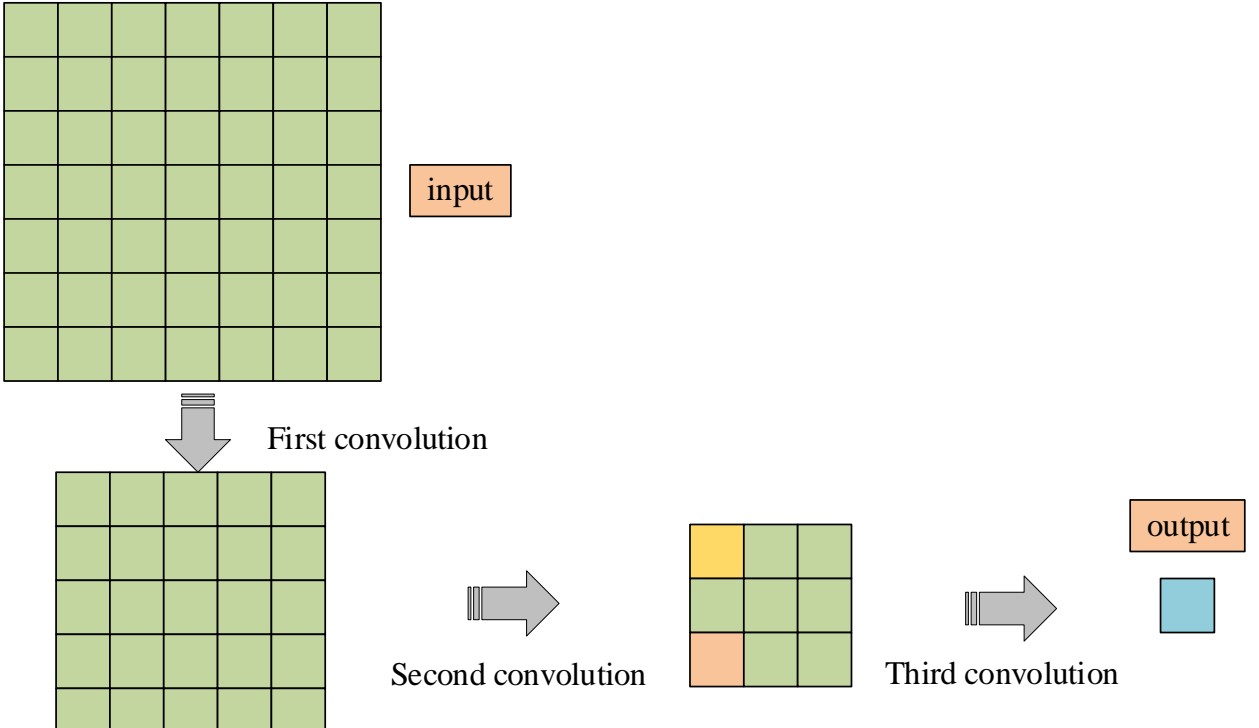

**Figure 5.** Receptive field.

The specific analysis of the calculation process of the receptive field is shown in Figure 5, the size of the image in the input network is $7 \times 7$, and the feature image of $1 \times 1$ is output after three convolutions, the step size of the convolution is 1, the padding is 0, and the size of the convolution kernel is $3 \times 3$. After n convolutions, the size of the receptive field of each unit in the feature image in the original image of the input network is recorded as $r_n$, the step size of the convolution is recorded as $s_n$, and the size of the convolution kernel is recorded as $k_n$. As shown in Figure 5, the size of the receptive field of the image after the first convolution on the original image is the same as the size of the convolution kernel; that is, $r_1 = 3$, and then the maximum receptive field of the image after the second convolution is $3 \times 3 = 9$ [28–30]. Since the step size of the first convolution is 1, the convolution kernels overlap during the moving process; thus, it is necessary to subtract the overlapping part $(k_2 - 1)$ from the feature image after the second convolution. The size of the receptive field of the overlapping part is $r_1 - s_1$, from which Equation (12) can be obtained:

$$r_2 = r_1 \times k_2 - (r_1 - s_1) \times (k_2 - 1) = 3 \times 3 - (3 - 1) \times (3 - 1) = 5 \tag{12}$$

Since the size of the convolutional receptive field is cumulatively affected by the step size of the previous convolution, Equation (13) can be obtained:

$$r_3 = r_2 \times k_3 - (r_2 - s_1 \times s_2) \times (k_3 - 1) = 5 \times 3 - (5 - 1 \times 1) \times (3 - 1) = 7 \tag{13}$$

By analogy, Equation (14) can be obtained:

$$r_n = r_{n-1} \times k_n - \left( r_{n-1} - \prod_{i=1}^{n-1} s_i \right) \times (k_n - 1), \, n \geq 2 \tag{14}$$

After simplifying the above equation, the specific calculation of the receptive field is obtained as shown in Equation (15).

$$r_n = r_{n-1} + (k_n - 1) \prod_{i=1}^{n-1} s_i, \, n \geq 2 \tag{15}$$

(2)    Dilated Convolutions

When making predictions on dense images, it is necessary to perform extensive reasoning on the context, and it is necessary to ensure that the obtained prediction results have high resolution. To solve the above problems, a dilated convolution algorithm is proposed, which can expand the size of the network's receptive field, and the image's resolution will not change during this process [31,32]. The one-dimensional calculation of dilated convolution is shown in Equation (16).

$$g[i] = \sum_{l=1}^{L} f[i + r.l]h[l] \tag{16}$$

In Equation (16), $g[i]$, $f[i]$ denote output and input signal; $h[l]$ refers to the convolution kernel of length $l$; $r$ means the dilated convolution coefficient used to sample the input signal $f[i]$. In the general convolution process, the dilated convolution coefficient is set to 1.

The expansion of the size of the receptive field of the convolution kernel in the semantic segmentation system can be achieved through dilated convolution of a two-dimensional form. The specific way is to insert a "hole" between two adjacent pixels of a common convolution kernel. In recent years, dilated convolutions have been widely used in semantic segmentation systems, which have effectively improved the accuracy of semantic segmentation [33,34].

*2.4. An ESNet for Semantic Segmentation*

To improve the accuracy of semantic segmentation, an ESNet for real-time semantic segmentation for autonomous unmanned system in traffic scenes is proposed. The results of the specific network are shown in Figure 6.

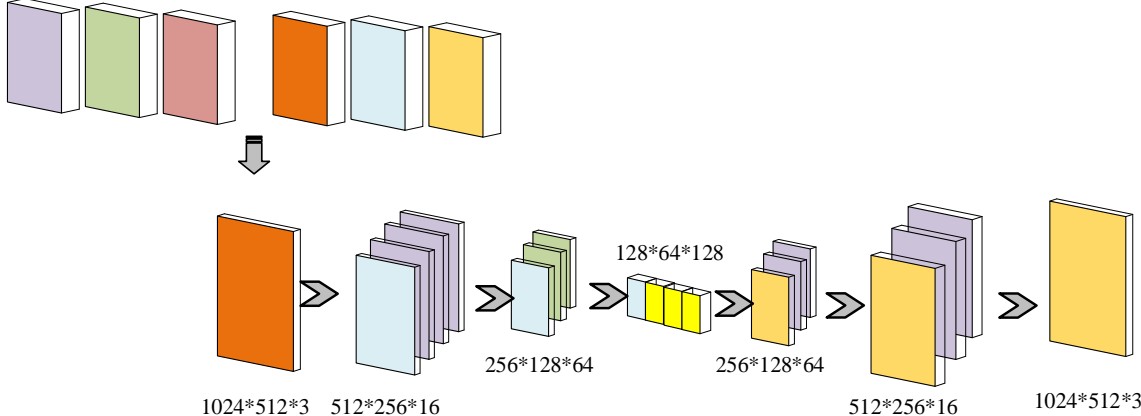

**Figure 6.** ESNet structure.

The difference between the ESNet network in this paper and the existing methods is mainly reflected in the network design [11–13]. As discussed in the introduction of this article, the basic framework of SegNet, a classical semantic segmentation network, adopts VGG16 deep learning network as an end-to-end network, which finally classifies images at the pixel level. ENet adopts an asymmetric encoder-decoder structure, effectively reducing the number of parameters. ESPNet uses an efficient spatial pyramid convolution module to improve the operation speed, but the accuracy and semantic segmentation effect need to be optimized. CGNet mainly relies on the context-guided (CG) block structure to improve the detection accuracy, which belongs to the network proposed after optimizing the local structure. ERFNet is similar to ENet, which is improved under the framework of the ResNet network. The operation of low-rank approximation significantly improves the calculation accuracy and leads to many parameter settings required by the network. ICNet adopts the structure of multi-scale prediction fusion and improves the classification accuracy by integrating the low-level semantic information and high-level detail information. Although the above networks are optimized in terms of detection accuracy and efficiency, most semantic

segmentation networks are challenging to achieve. Most optimization methods used by semantic segmentation networks mainly include designing high-performance processing structures, using the encoder–decoder structure, and multi-scale prediction fusion. It is not easy to achieve efficient and accurate semantic segmentation by focusing on only one of the optimization methods. Based on this, we propose an ESNet semantic segmentation network by integrating various optimization methods. The ESNet network adopts the above optimization methods, and the independently designed PFCU structure further improves the network's robustness and detection accuracy. Through some optimization design, the network has apparent advantages in the semantic segmentation of the urban environment. The structural design of ESNet network is introduced in detail below.

ESNet is an efficient symmetrical encoder–decoder structure consisting of four units: up-sampling, Parallel Factorized Convolution Unit (PFCU), Factorized Convolution Unit (FCU), and down-sampled. Among them, the residual module of the FCU adopts 1D factorized convolution with different sizes of convolution kernel, which enables it to capture object instances of different sizes. In order to effectively improve the feature expression ability of the network, the residual module of the PFCU adopts the strategy of "transform-split-transform-fusion". It uses convolution with different atrous coefficients to expand the receptive field of the convolutional layer. The ESNet structure is constructed by symmetrically stacking the residual modules of PFCU and FCU. The encoder generates the deep semantic feature image of the down-sampled unit. In contrast, the decoder up-samples the deep feature image to match the resolution of the input image so that the spatial information of the image is recovered and mapped to the segmentation category. The semantic segmentation prediction results are finally generated, and the obtained prediction results are consistent with the resolution of the input image [35,36]. The structure level of ESNet is shown in Figure 7.

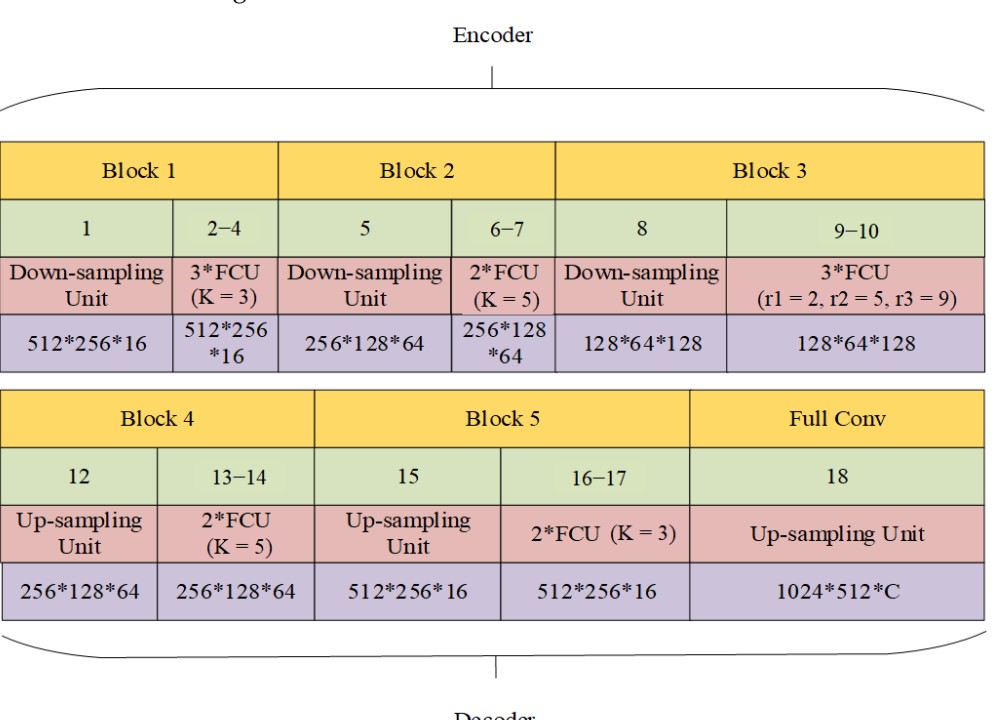

**Figure 7.** The structure level of ESNet.

(1)　Hybrid residual block of atrous factorized convolutional encode

The PFCU is the core component of the designed ESNet structure, and its specific structure is shown in Figure 8. The residual module of the PFCU adopts the strategy of "transform-split-transform-fusion". First, the data are processed from the $3 \times 1$ conv layer to the $1 \times 3$ conv layer through the Relu activation function. This process is the first

transform operation. Then, after the data pass through the BN layer, they are processed into three concurrent channels through the Relu activation function, which is the split operation. On this basis, the data of the three channels are processed by the Relu activation function respectively and then transferred from the 3 × 1 conv layer to the 1 × 3 conv layer. This process is the second transform operation. Finally, afterward, the data of the three channels are merged through the BN layer and processed by the Relu activation function for the last time, this process is the fusion operation. Considering the complexity of the actual urban environment, we try to adopt some robust means in designing the core processing structure of the ESNet network proposed in this paper. For example, using the BN layer here stabilizes the training and reduces the extreme distribution of input data caused by instability.

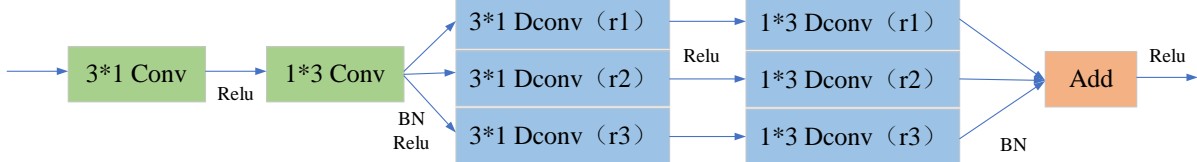

**Figure 8.** The structure of PFCU.

The specific structures of the bottleneck residual block, non-bottleneck residual block, one-dimensional non-bottleneck residual block and FCU in ResNet are shown in Figure 9.

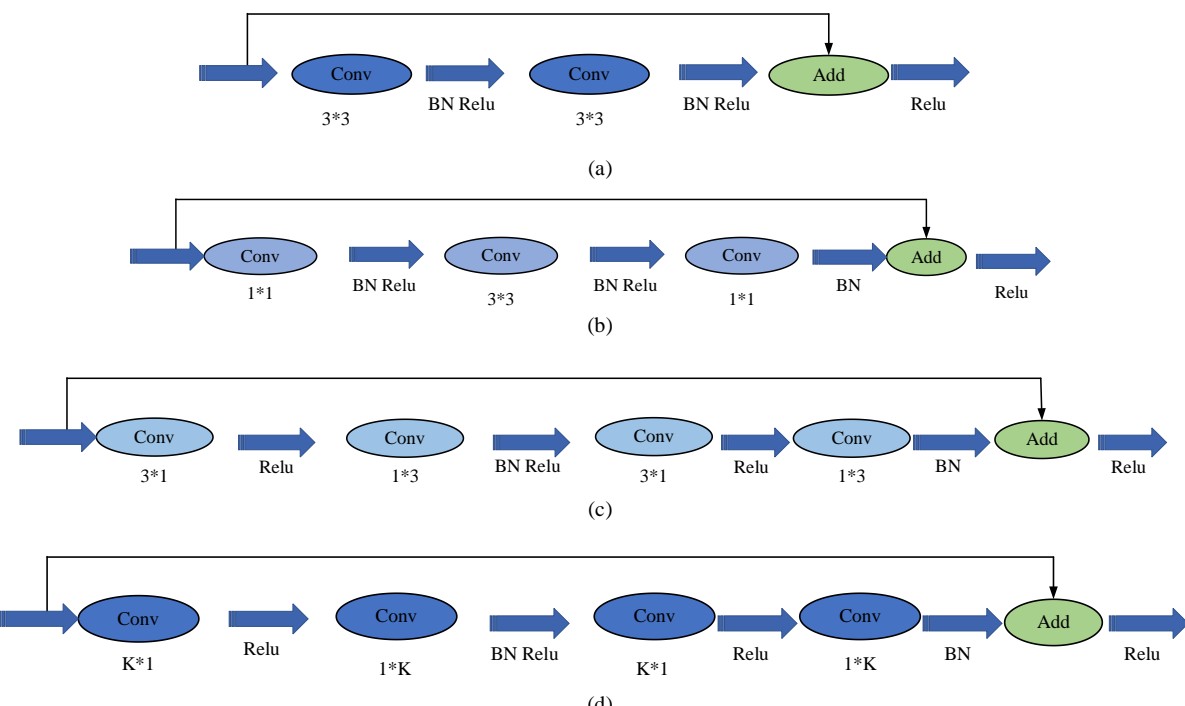

**Figure 9.** Structure diagram of the residual module of the PFCU: (**a**) non-bottleneck residual block; (**b**) bottleneck residual block; (**c**) one-dimensional non-bottleneck residual block; (**d**) FCU.

The size of the convolutional layer in the main branch structure of the bottleneck residual block is 1 × 1, 3 × 3, and 1 × 1. The function of two convolutional structures is to increase or decrease the channel dimension, and an activation function is added between the convolutional layers to realize batch processing of the normalization layer. Shortcuts connect the side branches, and the output of the main branch is directly added pixel by pixel.

The main branch structure of the non-bottleneck residual block consists of two convolutional layers with a size of 3 × 3; a Batch Normalization (BN), and a Rectified Linear

Unit (Relu) activation function are added between the two convolutional layers. Shortcuts connect the side branches, and the residual block's input and the main branch's output are directly added pixel by pixel. Problems such as gradient exploding and vanishing in the parameter update of CNN can be avoided by shortcut connection, and a deeper network can be trained.

(2)    Up-sampling module and down-sampled module

The specific structure of the up-sampling and down-sampled modules used is shown in Figure 10.

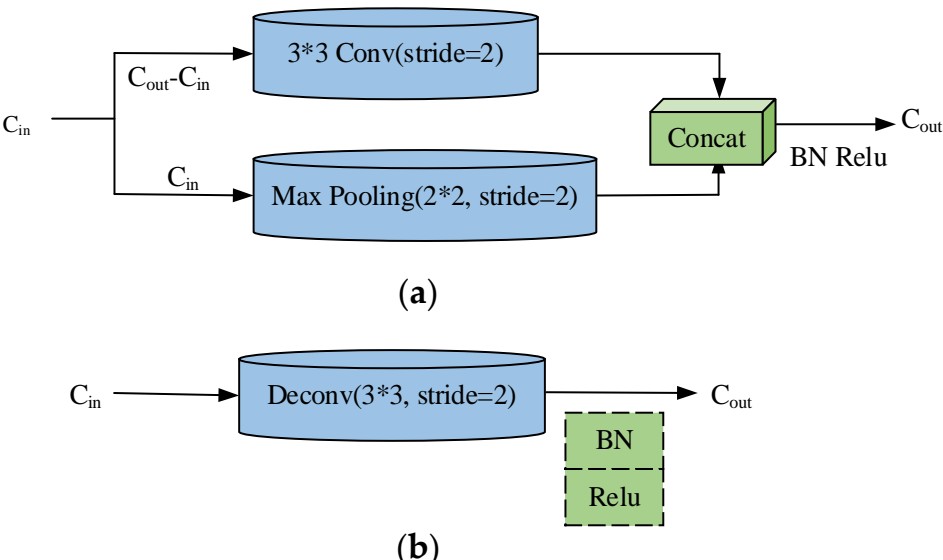

**Figure 10.** Structure of up-sampling module and down-sampled module: (**a**) up-sampling module; (**b**) down-sampled module.

The specific structure of the up-sampling and down-sampled modules used is shown in Figure 10, the size of the convolution kernel in the up-sampling module is $3 \times 3$, the step size is 2, and the input and output tensors are $(N, C_{in}, H_{in}, W_{in})$ and $(N, C_{out}, H_{out}, W_{out})$, respectively. The number of channels of the convolution kernel is $C_{out}$ and deconvolution with padding, so that the resolution of the image can be recovered. The number of input and output channels of the down-sampled module is $C_{in}$ and $C_{out}$, respectively. The structure adopted is a parallel branch design. The size of the convolution kernel in the main branch structure is $3 \times 3$, the step size is 2, and the number of channels is $C_{out} - C_{in}$. The number of branch output channels is the same as the number of channels of convolution kernel. The maximum confluence layer size in the side branch structure is $2 \times 2$ and the step size is 2. Then, the number of output channels of this layer structure is $C_{in}$. The feature image obtained by the branch are connected and used as the output feature image of the entire down-sampled module, and the number of channels is $C_{out}$.

## 3. Experiment and Results Analysis

### 3.1. Experimental Configurations

Pytorch is a Python ML library based on the Torch language. Torch is an open-source ML library based on the Lua programming language. Therefore, Pytorch has an automatic differentiation mechanism that can build and train neural networks. Moreover, it has tensor computing supported by potent Graphics Processing Unit (GPU) acceleration such as NumPy.

The Pytorch framework has been widely used in recent years because of its significant advantages. The Pytorch framework is different from DL libraries such as Tensorflow. Tensorflow requires that the complete calculation graph be defined before the user-defined

model can be run. However, the Pytorch framework supports the dynamic graph mechanism, and the Pytorch framework effectively improves the speed and flexibility of DL-related research.

Therefore, the Pytorch framework is used to analyze the performance of the proposed algorithm. The experimental configurations are shown in Figure 11.

| Software and hardware environment | operating system | GPU | GPU processor | CUDA | cuDNN | Deep learning framework |
|---|---|---|---|---|---|---|
| configuration | Ubuntu 16.04 | NVIDIA GeForce GTX 1080 Ti | Intel Xeon W-2133(3.60GHz) | CUDA 9.0 | cuDNN7.14 | Pytorch 1.1 |

**Figure 11.** The experimental configurations.

The datasets used in this experiment are Cityscapes and CamVid autonomous driving scene datasets:

(1) Cityscapes dataset: This dataset collects 50 different urban scene data, including vehicles, pedestrians, roads, etc. It contains an image set with 20,000 rough annotations and an image set with 5000 good annotations. The 5000 dataset finely labeled image sets are subdivided during the experiment to form 2975 training sets, 1525 testing sets, and 500 validation sets, with 19 target categories and 1 additional background category. After the segmentation is over, the semantic segmentation results are further improved using a set of images with rough annotations [37]. An example of image annotation in the Cityscapes dataset is shown in Figure 12.

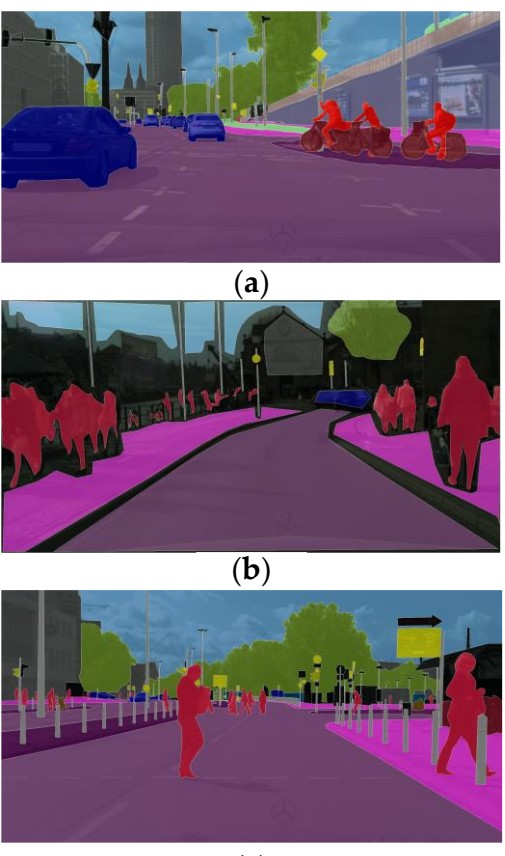

**Figure 12.** An example of image annotation from the Cityscapes dataset: (**a**) training set/test set—fine annotation; (**b**) training set—rough annotation; (**c**) test set—fine annotation.

(2) CamVid dataset: This dataset contains 710 low-resolution images with 11 segmentation categories. During training, the 710 low-resolution images are further divided into training, validation, and test sets with 376/101/233 images, respectively [38].

### 3.2. Experimental Configurations

The training process of the specific model is as follows:

First, the data are preprocessed, and the effectiveness of the proposed algorithm is verified by using the Cityscapes scene segmentation dataset. The dataset contains 5000 finely labeled images with a resolution of $1024 \times 2048$, including 2975 images in the training set and 1525 images in the test set, and 500 images in the validation set. Moreover, 20,000 roughly annotated images are used to improve the generalization performance of the designed segmentation network. The model is first trained with roughly labeled images. After 20–30 epochs of training, the model is continued to be trained with finely labeled images. Afterward, the images in the dataset are cropped to a size of $512 \times 1024$ and then input into the network. During the experiment, the data are enhanced using overall image translation and random horizontal flipping.

In the model training phase, the Adam optimizer is used to update the model parameters first. During the training process, the relationship between the training set Loss of the model, the learning rate, and the number of training rounds is shown in Figure 13.

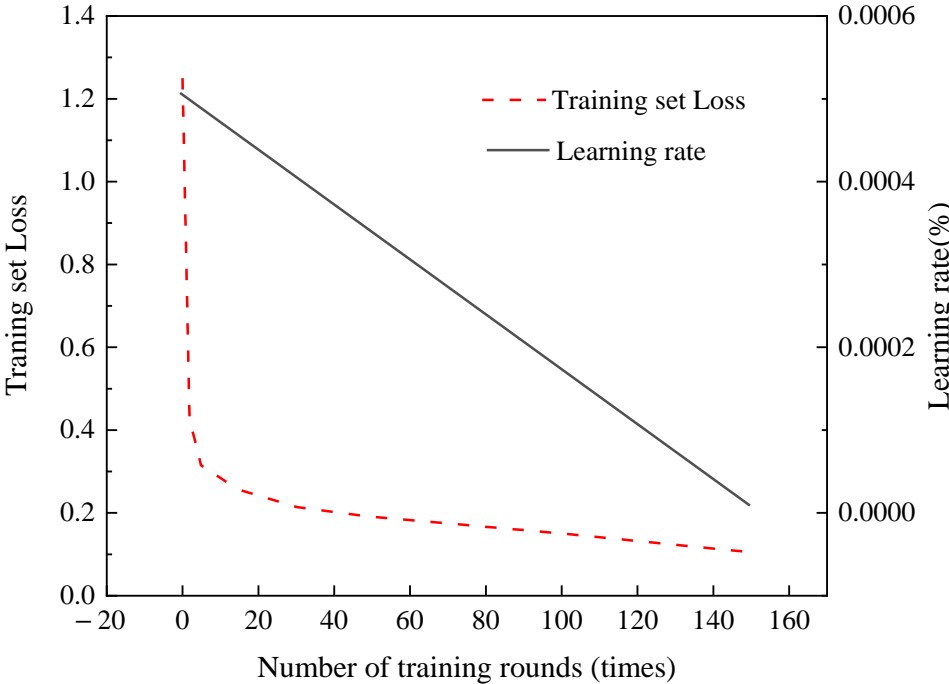

**Figure 13.** Curve graph of the training set Loss and the learning rate and the number of training epochs.

### 3.3. Comparison of Model Experiments

3.3.1. Experimental Results and Analysis on the Cityscapes Dataset

To verify the performance of the proposed ESNet for segmentation, six real-time semantic segmentation models with higher accuracy are selected for comparison with the proposed network model, namely SegNet, ENet, ESPNet, CGNet, ERFNet, and ICNet, etc. The experimental results obtained are shown in Figure 14.

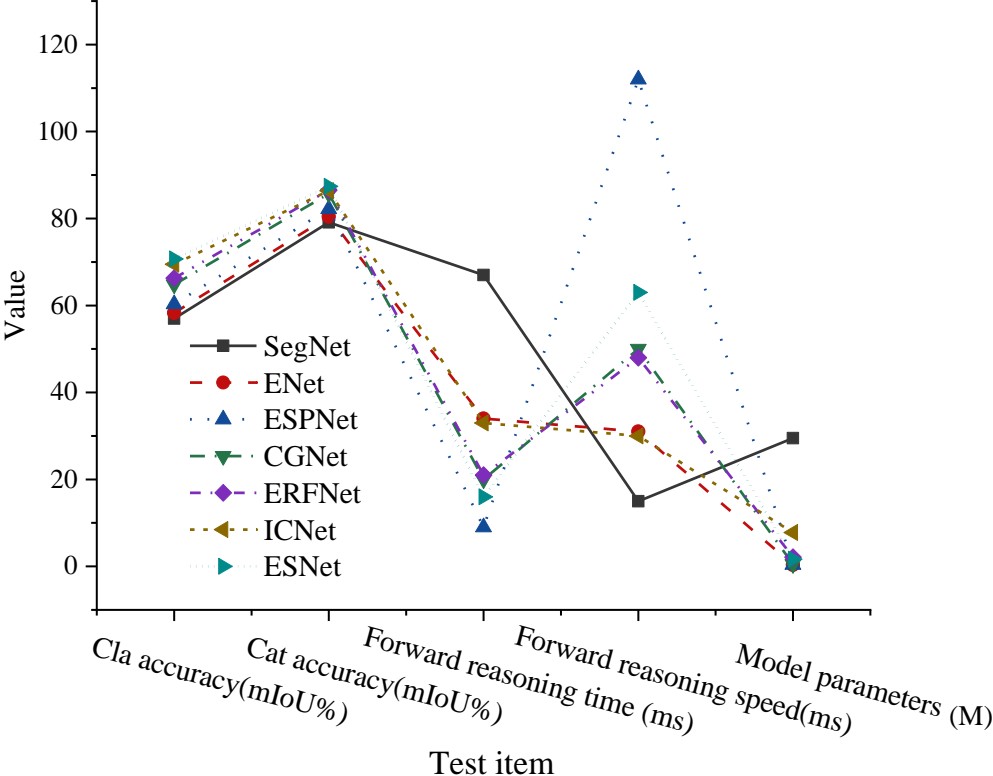

**Figure 14.** Comparison results of various segmentation methods.

The experimental results obtained are shown in Figure 14. The Cla accuracy refers to the average IoU of seven segmentation methods in 19 semantic categories, such as bicycle, motorcycle, train, bus, truck, car, rider, person, sky, terrain, vegetation, traffic signs, traffic lights, poles, fences, walls, buildings, sidewalks, roads, etc. Cat accuracy is the average IoU of seven segmentation methods under seven large grouping categories: people, objects, sky, vehicles, natural, buildings, planes, etc.

In Figure 14, on the Cityscapes dataset, the ESNet structure achieves a segmentation accuracy of 70.7% for the 19 semantic categories set and 87.4% for the seven large grouping categories. Compared with other algorithms, all have different degrees of increase. Compared with a complex ICNet, the segmentation accuracy of the proposed ESNet structure is only 0.4% lower than it in each semantic category. However, its segmentation accuracy is improved by 0.4% under large grouping categories. The recognition results of the seven segmentation models for semantic categories such as roads, sidewalks, buildings, walls, fences, and poles are shown in Figure 15.

Comparing the recognition results of the above 13 categories by different segmentation models indicates that the constructed ESNet structures have achieved better segmentation results. Compared with the ENet structure, the ESNet structure improves the segmentation accuracy of trucks by 22.9% and the segmentation accuracy of cars by 29.6%, which has obvious advantages. High recognition accuracy is still achieved for pixel categories that are difficult to identify, such as fences, poles, and traffic facilities. This demonstrates that the recognition accuracy of different network structures for small target objects such as traffic lights and traffic signs is different, but the ESNet network proposed in this paper can still better complete the task of semantic segmentation.

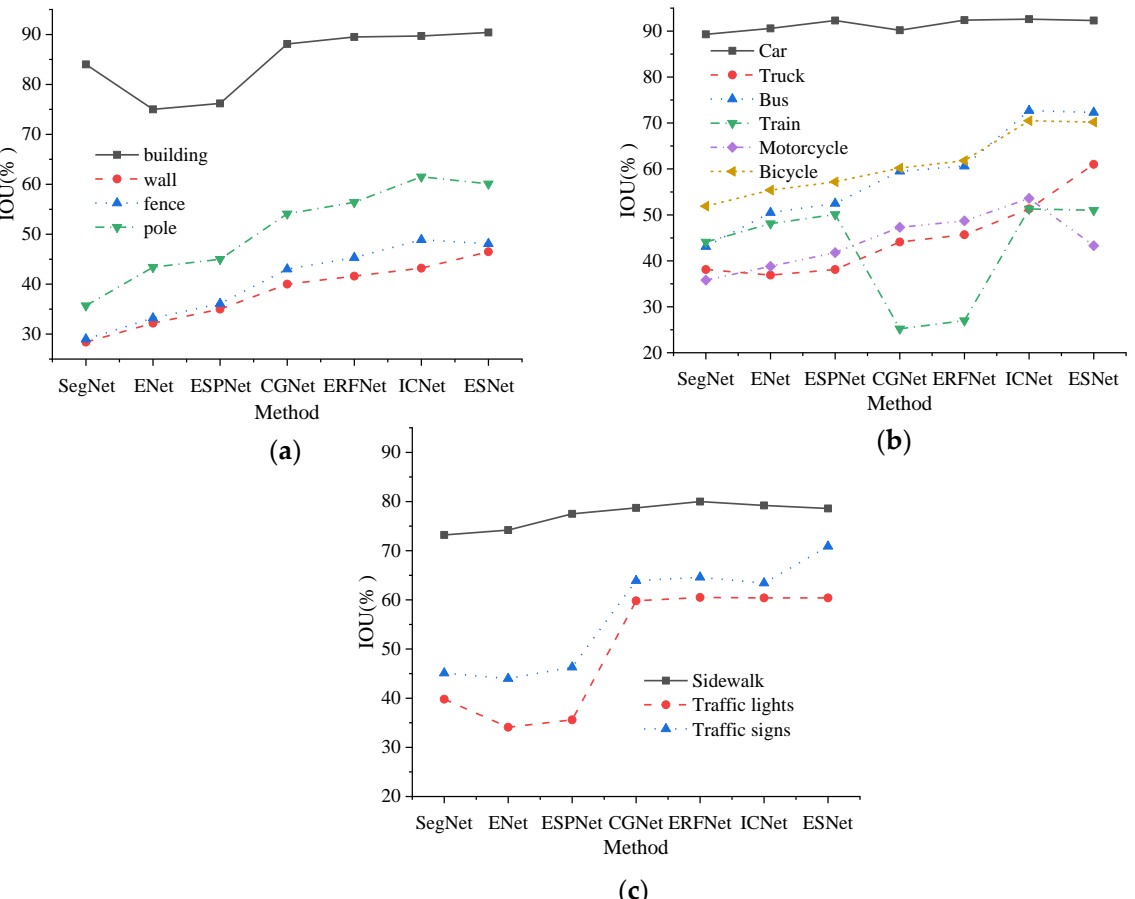

**Figure 15.** Comparison of recognition results of various segmentation models: (**a**) recognition results of various buildings; (**b**) recognition results of various traffic facilities; (**c**) recognition results of various traffic signs.

3.3.2. Experimental Results and Analysis on the CamVid Dataset

To verify the efficiency of the proposed ESNet, this section trains and evaluates the ESNet on the CamVid scene segmentation dataset and compares and analyzes it with several current fast segmentation methods. The evaluation indicators used include: Mean Intersection over Union (MIoU), model parameters, and Frames Per Second (FPS).

Compared with some current lightweight real-time semantic segmentation methods, the overall segmentation performance of the proposed ESNet structure on the CamVid test set is shown in Figure 16.

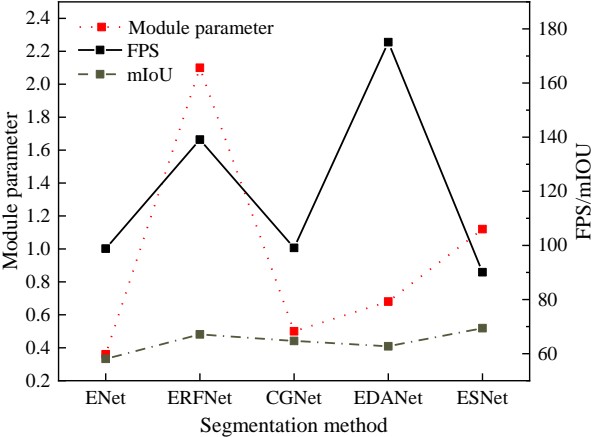

**Figure 16.** Segmentation comparison results of ESNet on the CamVid test dataset.

In Figure 16, the experimental results reveal that compared with segmentation networks of multiple lightweight real-time images, the parameters of the ESNet model are around 1.2 m, the highest FPS value is around 90 Hz, and the highest mIOU value is around 70%. It can be seen that ESNet significantly improves segmentation accuracy while maintaining faster forward inference speed.

To further analyze the prediction accuracy of the ESNet structure on each category, the prediction accuracy of ESNet on the CamVid test set for each category is shown in Figure 17.

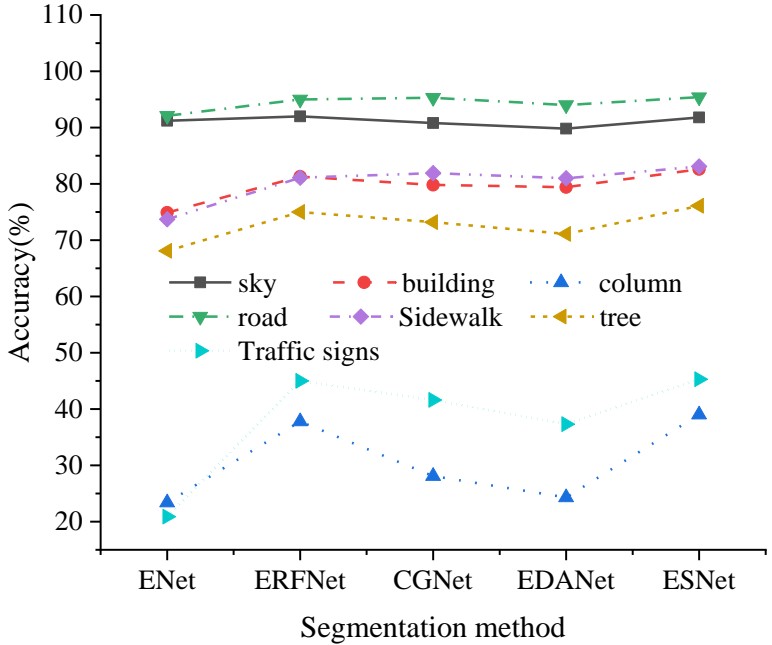

**Figure 17.** Prediction accuracy of ESNet on various categories.

Figure 17 demonstrates that in the seven semantic categories of the sky, buildings, pillars, roads, sidewalks, trees and traffic signs, the segmentation accuracy of the ESNet model is the highest at around 98% and the lowest at around 40%. In the segmentation test of each category, the ESNet model has the highest segmentation accuracy. This means that the performance of the ESNet model is better than other models.

### 3.3.3. Efficiency of Scene Semantic Segmentation in Different Networks

In order to better test the processing ability of the method proposed in this paper in the actual urban scene, we use the real image collected by the front camera of the vehicle to test. Three pieces of test data in three different scenarios are mainly used. Finally, the average value of the shortest time taken to obtain reliable test results is counted. The test results are shown in Table 1 in milliseconds.

**Table 1.** Efficiency comparison table of different methods (ms).

|  | SegNet | ENet | ESPNet | CGNet | ERFNet | ICNet | EDANet | ESNet |
|---|---|---|---|---|---|---|---|---|
| Daytime scene | 67.5 | 38.6 | 9.6 | 25.1 | 25.7 | 40.0 | 21.2 | 20.8 |
| Night scene | 43.2 | 30.0 | 9.1 | 21.6 | 24.5 | 37.2 | 20.8 | 19.6 |
| In rainy weather | 60.0 | 33.8 | 9.4 | 24.5 | 23.5 | 38.6 | 21.6 | 20.4 |
| Average value | 56.9 | 34.1 | 9.3 | 23.8 | 24.6 | 38.6 | 21.2 | 20.3 |

From the experimental results, the method in this paper performs well in the actual tests in many scenarios. Although ESNet's average detection speed ranks second, its detection accuracy is more than 15% higher than that of the fastest ESPNet network. In general, this method has apparent efficiency and accuracy advantages in dealing with semantic

segmentation in the complex urban environment. In other words, the ESNet network considers these two aspects simultaneously and has excellent practical application value.

### 3.4. Discussion

According to DL theory, the method for efficient image semantic segmentation using DCNN is deeply studied. This is mainly achieved by optimizing DCNN for efficient semantic segmentation of images and applying it to vehicle visual sensors to improve its recognition performance for urban scenes, thereby promoting the development of urban traffic. In the experiment of feature semantic segmentation using the Cityscapes dataset, ESNet has the highest Cla and Cat accuracy compared with other networks and is at a high level in indicators such as forward inference time and inference speed. At the same time, the training of the model requires the fewest parameters. Moreover, the IOU of ESNet in extracting static objects, moving objects, and small targets is basically at a high level in each network, and some projects are better than all other networks, which also show the advantages proposed by ESNet. It mainly studies the semantic segmentation of multi-class objects in complex urban environments. In experiments using the CamVid dataset, ESNet achieves the highest mIOU index compared to the other four existing semantic segmentation networks. Except for the Cityscapes dataset, the network still maintains competitive performance. In particular, whether using the Cityscapes dataset or the CamVid dataset, the semantic segmentation effect of the ESNet on small objects such as traffic signs is better than all other networks participating in the comparative experiments. This indicates that ESNet has a better semantic segmentation effect for small objects in the urban environment.

To sum up, ESNet has shown excellent results in both the accuracy and computational efficiency of semantic segmentation. Tested on different urban datasets, ESNet performs well on the task of semantic segmentation with various urban feature elements compared to other similar networks. The limitations of the research include the following two points: (1) Although DCNN is deeply optimized for efficient image semantic segmentation, it has not been practically applied; thus, there is a lack of evaluation results in practical traffic applications. (2) The reference factors for the optimization of semantic segmentation are not comprehensive enough; thus, more comprehensive optimization of this technology is needed in future research.

### 4. Conclusions

With the progress of AI technology, DL networks and optimization strategies for semantic segmentation have also been proposed in various industries. However, this technology has not been applied in urban traffic management. Therefore, based on DL theory, methods for efficient semantic segmentation of images using DCNN have been intensively studied. First, the theoretical basis of CNN is briefly stated, and the real-time semantic segmentation technology of urban scenes based on DCNN is introduced in detail. Then, the atrous convolution algorithm and the multi-scale parallel atrous spatial pyramid model are expounded, and an ESNet of real-time semantic segmentation model for autonomous driving scenarios is proposed. The overall network adopts an encoder–decoder structure. The encoder uses a designed hybrid atrous convolution module to extract dense image features and uses convolution kernels of diverse sizes at various levels of the network to extract multi-scale features. A large number of comparative experiments are designed to verify the effectiveness of the network, and the segmentation performance of the model is evaluated on the urban scene dataset Cityscapes. The proposed ESNet structure achieves a segmentation accuracy of 70.7% for the 19 semantic categories and 87.4% for the seven large grouping categories, which are increased to varying degrees compared with other algorithms, and this structure improves the segmentation accuracy of trucks by 22.9% and the segmentation accuracy of cars by 29.6%, which has obvious advantages. Although the proposed method achieves a balance between segmentation accuracy and forward reasoning speed to a certain extent, there are still many shortcomings that need to

be further studied and improved. The proposed efficient real-time semantic segmentation method based on DCNN faces huge challenges. Especially with the development of DL technology, many efficient methods have emerged. For example, the Generative Adversarial Networks can be used to learn more discriminative features, Graph network models are used to perform better relational reasoning on the data, NN architecture search is used to reduce manual operations in the design of the model, and the optimal network structure for fitting the data are adaptively selected. The network structure further reduces the memory consumption, storage overhead, and energy overhead of the model. These can further reduce the memory consumption, storage overhead, and energy costs of the model. Compared with the study of Wen et al. (2019) [39], the designed method covers a wider range of fields, and the degree of method optimization is deeper. Therefore, the research will play a more vital role in future social development.

**Author Contributions:** Conceptualization, Y.L. (Yanyi Li) and J.S.; methodology, Y.L. (Yanyi Li); software, Y.L. (Yuping Li); validation, Y.L. (Yanyi Li), J.S., and Y.L. (Yuping Li); formal analysis, Y.L. (Yanyi Li); investigation, J.S.; resources, J.S.; data curation, Y.L. (Yuping Li); writing—original draft preparation, Y.L. (Yanyi Li); writing—review and editing, J.S.; visualization, Y.L. (Yuping Li); project administration, J.S.; funding acquisition, J.S. All authors have read and agreed to the published version of the manuscript.

**Funding:** This work was supported by the National Innovation Training Program for College Students (No. 202010424004) and Frontier Physics Experiment Facility for Extremely Deep Underground Extremely Low Radiation Background of Key National Technology Infrastructure Construction-Subproject of pre-research project (Total No. 2018-000052-73-01-002125).

**Institutional Review Board Statement:** Not applicable.

**Informed Consent Statement:** Not applicable.

**Data Availability Statement:** The data involved in this paper can be obtained by contacting the corresponding author. In addition, this study involves two types of datasets. Please refer to the website for details of the Cityscapes dataset: https://www.cityscapes-dataset.com (accessed on 20 May 2022). For more information on another CamVid dataset, please refer to the website: http://mi.eng.cam.ac.uk/research/projects/VideoRec/CamVid/ (accessed on 21 June 2022).

**Acknowledgments:** Y.L. (Yanyi Li), Y.L. (Yuping Li), and S.J. carried out the calculation, result analysis, and drafted the manuscript, which was revised by all authors. All authors gave their approval of the version submitted for publication.

**Conflicts of Interest:** The authors declare no conflict of interest.

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
