# Peer review of "Real-Time Semantic Understanding and Segmentation of Urban Scenes for Vehicle Visual Sensors by Optimized DCNN Algorithm"

_applsci, doi:10.3390/app12157811_

Round 1

Reviewer 1 Report

This paper proposed Real-time Semantic Understanding and Segmentation of Urban Scenes for Vehicle Visual Sensors by Optimized DCNNs Algorithm. There are crucial problems that need to be carefully addressed as below:

1. The whole abstract should be rewrite again to be concise and clear more.

2. A deep literature review should be given, particularly regarding state-of-the-art machine learning methods such as:

Flood Mapping Using Relevance Vector Machine and SAR Data: A Case Study from Aqqala, Iran (10.1007/s12524-020-01155-y)

Development of a method for flood detection based on Sentinel-1 images and classifier algorithms (10.1111/wej.12681)  

3. The introduction section is weak; it didn't well provide information needed for the study.

4. The differences between the proposed method and existing methods is not clear. Please explain.
5. The experiments should be given more to show the effectiveness of the proposed method.

6. Result section needs more details.

7. Please add the limitation of the study in discussion section.

8. The Conclusion is so simple and it is repeated words from the manuscript, the Conclusion need to rearrange in different way from the abstract and to be clear and concise.

Reviewer 2 Report

The authors use deep learning approaches to segment Urban from Cityscapes and CamVid images. They showed the ESNet model achieved higher accuracy while the model speed increased. Comparison to the other segmentation models performed well. The quality of the manuscript is very good. 

Author Response

Dear reviewer,

Thank you for your letter and the reviewers’ comments on our manuscript. Those comments are very helpful for revising and improving our paper and the important guiding significance to other research. 

Once again, thank you very much for your constructive comments and suggestions.

Best regards,

Li Yanyi

Reviewer 3 Report

Comments

1 – Acronyms have no plural (on the title and on the entire paper, please change DCNNs -> DCNN)

2 – The abstract is a bit long. Please, provide a shorter and more concise version of the abstract with the key ideas about the work.

3 – At the end of section 1, Line 82, please describe the organization of the paper, please state the number of the remaining sections along with their contents.

4 – Equation (1). Please check the symbols. On equation (1) we have the symbol ‘b’. After equation (1), we have ‘B’.

5 – Please treat equations as elements of the text. The punctuation rules also apply to equations. In many equations, there is a period missing.

6 – In many places in the manuscript, we have “…is shown in Figure ##:”, where ## is the figure number. Please change to “…is shown in Figure ##.”

7 – Equation (2) defines the activation function as f(x). On Equation (3) to Equation (7), the activation function is defined as ‘y’. Please check this and use a consistent notation for the activation function.

8 - Section 3.1. Please add a proper reference to the Cityscapes and CamVid datasets.

9 – From Lines 354 and 359, we have repetition of text regarding the Cityscapes dataset description.

This dataset collects 50 different urban scene data, including vehicles, pedestrians, roads, etc. It contains an image set with 20k rough annotations and an image set with 5k good annotations. The dataset used in this experiment is the Cityscapes scene segmentation dataset, which collects 50 different urban scene data, including vehicles, pedestrians, roads, etc. It includes an image set with 20k rough annotations and an image set with a set of 5k finely annotated images.

10 – Line 365, we have 701 low-resolution images. On Line 367, we have “Further divided into training, validation, and test sets with 376/101/233 images”. Notice that 376 + 101 + 233 = 710. Please check.

11 – Figure 14. On the yy label of the graphic, please put a description on the measures [%] and [ms] to be clearer. On the xx label, please state what is the meaning of “Model parameters” metric?

12 - Figure 16. Regarding the 3 metrics, we have “Module parameter”, “FPS”, and “mIoU”, please state the corresponding yy axis.

13 – Figure 17. On the yy label of the graphic, please put a description on the considered metric.

Writing

Line 33

artificial intelligence (AI)..

->

artificial intelligence (AI).

Line 124

The fully connected layer is neurons of a single layer,

->

The fully connected layer is composed by neurons on a single layer,

Line 139

The image of the activation function is shown in Figure 4:

->

The plot of these activation functions is shown in Figure 4.

Line 141

Figure 4. The image of the activation function.

->

Figure 4. The plots of the activation functions.

Lines 211 and 212

in autonomous driving, etc. need to be carefully evaluated.

->

in autonomous driving, need to be carefully evaluated.

Line 327

Cin and Cout respectively.

->

Cin and Cout, respectively.

Line 331

Then the number

->

Then, the number

Line 426

and traffic signals

->

and traffic signs

Round 2

Reviewer 1 Report

The paper is indeed interesting but it needs further scientific editing.

  1. Some important numerical findings should be provided in the Abstract.
  2. The logical flow in the Introduction is not well developed and it is not clear for the readers what are the key knowledge gaps and objectives of the researchers to fill them.
  3. Please be clear about the purpose of the article and the research innovation aspects at the end of the introduction.
  4. The differences between the proposed method and existing methods are not clear.
  5. The contributions of this paper are not so clear to the reviewer.
  6. The proposed method suggests adding more details.
  7. The experiments should be given more to show the effectiveness of the proposed method.
